# Identification of the Genes Encoding B3 Domain-Containing Proteins Related to Vernalization of *Beta vulgaris*

**DOI:** 10.3390/genes13122217

**Published:** 2022-11-25

**Authors:** Naiguo Liang, Dayou Cheng, Li Zhao, Hedong Lu, Lei Xu, Yanhong Bi

**Affiliations:** 1School of Life Sciences and Food Engineering, Huaiyin Institute of Technology, Huaian 223001, China; 2School of Chemical Engineering and Technology, Harbin Institute of Technology, Harbin 150001, China

**Keywords:** sugar beet, whole-transcriptome sequencing, vernalization, B3 domain-containing proteins, lncRNA

## Abstract

Vernalization is the process of exposure to low temperatures, which is crucial for the transition from vegetative to reproductive growth of plants. In this study, the global landscape vernalization-related mRNAs and long noncoding RNAs (lncRNAs) were identified in *Beta vulgaris*. A total of 22,159 differentially expressed mRNAs and 4418 differentially expressed lncRNAs were uncovered between the vernalized and nonvernalized samples. Various regulatory proteins, such as zinc finger CCCH domain-containing proteins, F-box proteins, flowering-time-related proteins FY and FPA, PHD finger protein EHD3 and B3 domain proteins were identified. Intriguingly, a novel vernalization-related lncRNA–mRNA target-gene co-expression regulatory network and the candidate vernalization genes, *VRN1*, *VRN1-like*, *VAL1* and *VAL2*, encoding B3 domain-containing proteins were also unveiled. The results of this study pave the way for further illumination of the molecular mechanisms underlying the vernalization of *B. vulgaris*.

## 1. Introduction

The transition from vegetative to reproductive growth represents the predominant developmental stage for terrestrial plants. This transformation is an output response to various external stimuli and internal cues that later results in flowering. Five genetic pathways were found to control the floral transition involved in the vernalization pathway, photoperiod pathway, the gibberellin pathway, the autonomous pathway and the plant age, among which the vernalization exposure to a long period of cold played a crucial role in the floral initiation [1]. In recent years, various vernalization-related genes have been explored in plants by RNA-Seq. For instance, two vernalization-related genes (LoSVP and LoVRN1) were identified to play key roles in the vernalization pathway of oriental lily by RNA-Seq [2]. The association of WRKY, NAC, AP2/EREBP, AUX/IAA, MADS-BOX, ABI3/VP1, bHLH and CCAAT family transcription factors with circadian rhythm was revealed in vernalized orchardgrass by using high-throughput sequencing [3]. The bHLH, ERF and WRKY transcription factors involved in the vernalization response of faba bean were also explored by RNA-Seq [4]. Similarly, a series of vernalization-related candidate genes including *TPS*, *UGP*, *CDF* and *VIN1* were screened according to their expression patterns during vernalization by RNA-Seq [5].

The molecular mechanism of vernalization in *Arabidopsis thaliana* (*L*.) has been well uncovered, especially the *FLOWERING LOCUS C* (*FLC*). *FLC* was found to be the central modulator of the induction of flowering during vernalization, which encoded a MADS-box protein inhibiting the floral organ transition. Vernalization induces the developmental phase, which is mitotically stable, indicating an epigenetic basis behind it. *VERNALIZATION1* (*VRN1*) and *VERNALIZATION2* (*VRN2*) were identified in the vernalization pathway of *A. thaliana*, where *VRN1* encoded a protein targeting the floral repressor *FLC*, and *VRN2* attenuated *FLC* levels by encoding a nuclear localized zinc finger protein, with similarity to Polycomb group (PcG) proteins [6]. Studies demonstrated that *VRN2* loci may function on *FLC* chromatin remodeling during vernalization [6]. The vernalization-induced PHD finger protein distribution contributed to establishing the PHD–PRC2 complex that broadly recruited H3K27me3 in *FLC* chromatin to an extent sufficient for *FLC* stable silencing [7,8,9,10]. The particular trimethyl-lysine 27 (K27me3) methylation was enriched at the start of the *FLC* loci in response to vernalization [11]. In addition to mRNAs and the PHD–PRC2 proteins complex, long noncoding RNAs also play important roles in the regulation of *FLC* gene expression. It was demonstrated that multiple lncRNAs, including *COOLAIR*, *COLD ASSISTED INTRONIC NONCODING RNA* (*COLDAIR*) and *COLDWRAP* regulated the *FLC* levels via enrichment of H3K27me3 during vernalization [12,13,14,15]. *COOLAIR* as an *FLC* antisense transcript was found to physically interact with the *FLC* loci and accelerate transcriptional repression of *FLC*. In contrast, *COLDAIR* and *COLDWRAP* directly interacted with PRC2 to suppress *FLC* expression levels during vernalization [16]. A previous study also found that FCA protein binding of *COOLAIR* and SSU72 was essential for PRC2 enrichment and H3K27me3 deposition at *FLC* loci during vernalization in *A. thaliana* [17].

Sugar beet (*B. vulgaris L.*) is one of the most important industrial crops, accounting for 30% of the world’s sugar production. The *BvFL1* homolog of *FLC* in sugar beet was identified by phylogenetic analysis, i.e., downregulation in response to vernalization [18]. Furthermore, the associations of nucleotide polymorphisms in *BvFL1* with bolting before vernalization was also screened by EcoTILLING [19]. A RAV1-like AP2/B3 domain protein of sugar beet in response to vernalization was uncovered by RNA-Seq [20]. An aquaporin *BvCOLD1* gene contribution to cold tolerance of sugar beet was also found [21]. Moreover, two predominant pathways of protein processing of sugar beet association with vernalization were also uncovered by our previous research [22]. The whole-transcriptome RNA-seq was used to explore differentially expressed mRNAs and lncRNAs association with salt stress of sugar beet [23]. However, the vernalization-related genes, especially the long noncoding RNAs of sugar beet had not been comprehensive identified and characterized. Hence, it was very necessary to further investigate the vernalization genes and regulatory network to better understand the molecular mechanism of the vernalization of sugar beet.

In our research, whole-transcriptome sequencing was employed to the vernalized and nonvernalized sugar beet. A total of 570 mRNAs and 292 lncRNAs were identified, and the significantly expressed RNAs were also analyzed between the vernalized and nonvernalized samples. We found the first vernalization-related lncRNA–mRNA target-gene co-expression regulatory network in *B. vulgaris*, comprising three mRNAs *BvVAL1-like*, *Bv8_189990_perh.t1* and *Bv8_189991z13909_ihku.t1* and one lncRNA *MSTRG.26204*. Meanwhile, B3 domain-containing proteins VRN1, VRN1-like, VAL1 and VAL2 were possibly the vernalization genes involved in the response to cold treatment. Our findings further highlighted the underlying molecular mechanism of vernalization of *B. vulgaris*.

## 2. Materials and Methods

### 2.1. Plant Materials and Growth Conditions

The cultivated sugar beet line DY14-O was used as the plant materials throughout this study. The fifty root samples, almost equal in size, were treated with vernalization (4–5 °C) under darkness for 10 weeks, and leaves were collected for whole-transcriptome sequencing. The corresponding nonvernalized samples were used for the controls; meanwhile, the three biological and technical replicates were set. Leaves were harvested from the vernalized samples at five time points: two weeks of vernalization (vernalization-2W), four weeks of vernalization (vernalization-4W), six weeks of vernalization (vernalization-6W), eight weeks of vernalization (vernalization-8W) and ten weeks of vernalization (vernalization-10W) and the corresponding time points for the nonvernalized samples.

### 2.2. Library Construction and Sequencing

RNA was extracted from leaves of sugar beet by (MagMAX™-96) total RNA isolation kit (Invitrogen, MagMAX™-96, Waltham, MA, USA) following the manufacturer’s procedure. The total RNA concentration and purity of each sample were evaluated by a NanoDrop ND-1000 (NanoDrop, Wilmington, DE, USA) with RIN number > 7.0. Ribosomal RNA of total RNA were depleted, and then fragmented into small pieces to construct the final cDNA library for sequencing on an Illumina Hiseq 4000.

### 2.3. Transcripts Assembly and LncRNA Identification

Raw data were processed by cutadapt software (https://cutadapt.readthedocs.io/en/stable/ (accessed on 10 August 2022)) to remove the reads containing adaptor contamination. Then, the sequence quality was verified via FastQC (http://www.bioinformatics.babraham.ac.uk/projects/fastqc/ (accessed on 10 August 2022)). The reads were mapped to the genome of *B. vulgaris* by Bowtie2 and topaht2 [24,25]. The StringTie was employed to assemble the mapped reads of samples [26]. Then, a comprehensive transcriptome was reconstructed by Perl scripts, and then StringTie and Ballgown was employed to calculate the expression levels of all transcripts [26,27].

### 2.4. LncRNA Identification

First of all, transcripts that overlapped with known mRNAs and transcripts less than 200 bp were abandoned. CPC and CNCI were used to predict the transcripts with coding potential [28]. The CPC score < −1 and CNCI score < 0 of transcripts were removed. The remaining transcripts were the lncRNAs.

### 2.5. Different Expression Analysis of mRNAs and lncRNAs

FPKM of mRNAs and lncRNAs were calculated by StringTie [29]. The mRNAs and lncRNAs with log2 (fold change) > 1 or log2 (fold change) < −1 and with statistical significance (*p* value < 0.05) were selected by R package Ballgown [27].

### 2.6. Target Gene Prediction and Functional Analysis of lncRNAs

The cis-target genes of lncRNAs were predicted to further explore the function of them. LncRNAs may play a cis role in acting on neighboring target genes. In this study, BLAST2GO was used to analyze the target gene’s function of lncRNAs with significance (*p* value < 0.05) [30]. Gene ontology (GO) enrichment analysis of genes were conducted by GOseq R packages. Pathways of genes were demonstrated by the Kyoto Encyclopedia of Genes and Genomes (KEGG, http://www.kegg.jp (accessed on 10 August 2022)). KOBAS 2.0 (http://kobas.cbi.pku.edu.cn/download.do (accessed on 10 August 2022)) software was used to calculate the statistical enrichment of genes in the KEGG pathways.

### 2.7. Phylogenetic Analysis of Target Genes in B. vulgaris

The entire primary sequence of *A. thaliana AtVAL1*(AT2G30470) and *AtVRN1* (AF289051 and AF289052) was used for aligning with the target genes sequence, and then a phylogenetic tree was constructed with the software MEGA 11.0 (http://www.megasoftware.net (accessed on 15 August 2022)).

### 2.8. RNA Extraction, cDNA Synthesis and qRT-PCR

Leaves from five roots samples were pooled to form a single sample for total RNA isolation and purification. cDNA was synthesized with 3 mg of RNA using SuperScript IV Reverse Transcriptase (Invitrogen) according to manufacturer’s instructions. The qRT-PCR assays were conducted in triplicate. The qRT-PCR was performed by the SYBR Green PCR Mastermix Kit (Takara, San Jose, CA, USA) with a final reaction volume of 20 μL. Primers were designed and listed in Appendix A. Reactions were run in a ABI 7500 PCR with the following cycling protocol: 95 °C for 10 min, followed by 40 amplification cycles at 95 °C for 15 s and 60 °C for 1 min. *BvICDH* (F: CACACCAGATGAAGGCCGT; R: CCCTGAAGACCGTGCCAT) was the endogenous control gene for mRNAs and lncRNAs.

## 3. Results

### 3.1. Identification and Analysis of Differentially Expressed mRNAs between Vernalized and Nonvernalized Samples

We finally obtained 96,327,758 and 84,994,242 raw reads and 95,245,446 and 83,904,532 valid reads from the vernalized and nonvernalized samples, respectively (Appendix A). After filtering, the valid reads were mapped to the reference genome (RefBeet-1.1) of *B. vulgaris*. In total, 31,312 mRNAs were identified, among which 22,159 were differentially expressed in the vernalized samples and 9153 in the nonvernalized ones (Appendix A, Figure 1A). The significantly expressed mRNAs was screened, among which 474 were upregulated (83.16%) and 96 were downregulated (16.84%) (Appendix A, Figure 1B). The expression patterns of the differentially expressed mRNAs in each sample were exhibited by a heat map (Figure 1C). GO and KEGG enrichment analysis were also performed to explore the potential functions of these mRNAs.

A total of 563 GO terms were screened by GO enrichment analysis (Appendix A). The majority of the differentially expressed mRNAs were annotated to “regulation of transcription, DNA-binding”, “protein phosphorylation”, and “oxidation-reduction process” in the BP category, and to “nucleus”, “integral component of membrane”, and “membrance” in the CC category, while the GO terms were annotated to “protein binding”, and “ATP binding” in the MF category (Figure 1D). A total of 423 GO terms with significance (*p*-values ≤ 0.05) were obtained. These terms included the oxidation–reduction process, heme binding, extracellular region, apoplast, peroxidase activity, cell wall, response to oxidative stress, glucosinolate biosynthetic process, monolayer-surrounded lipid storage body, and response to light stimulus (Figure 1E) that may be involved in the vernalization pathway of sugar beet. KEGG enrichment analysis indicated that these differentially expressed mRNAs were assigned to 105 pathways, among which 6 metabolic pathways with significance (*p* ≤ 0.05) were included (Figure 1F) (Appendix A). The pathways “RNA transport”, “Homologous recombination”, “mRNA surveillance pathway”, “Base excision repair”, “Valine, leucine and isoleucine biosynthesis”, and “Spliceosome” were significantly enriched in vernalized samples.

### 3.2. Identification and Analysis of Differentially Expressed lncRNAs between the Vernalized and Nonvernalized Samples

In addition to mRNAs, 7132 lncRNAs were explored altogether, among which 4418 were upregulated in the vernalized samples, and 2714 were downregulated (Appendix A, Figure 2A). A total of 292 lncRNAs with (| log2 (fold change) | > 1 and *p* ≤ 0.05) were screened between vernalized and nonvernalized samples, among which 207 lncRNAs were upregulated (70.89%), and 85 were downregulated (29.11%) (Figure 2B). The general expression trends of these lncRNAs are shown in Figure 2C. Genomic comparisons between mRNAs and lncRNAs showed that long (>300 bp) mRNAs were more abundant than lncRNAs, whereas short (≤300 bp) mRNAs were less abundant than lncRNAs (Figure 2D). The open reading frame (ORF) of mRNAs was longer than that of lncRNAs. The length of the ORFs of the lncRNAs ranged from 0 to 500 aa, whereas the length of the ORFs of the mRNAs ranged from 0 to 2000 aa (Figure 2E,F). Furthermore, the exons (1–2) of differentially expressed lncRNAs were fewer in number than that of mRNAs on average, and the expression patterns of lncRNAs were lower than that of mRNAs (Figure 2G). To further investigate the functions of the lncRNAs, we also performed GO and KEGG enrichment analysis of the genes targeted by lncRNAs (Appendix A). The targets of the lncRNAs were assigned to 124 GO terms, including “oxidation–reduction process”, “transmembrane transport”, “carbohydrate metabolic process”, and “metabolic process”, under BP; “membrane”, “nucleus”, “cytoplasm”, “integral component of membrane”, and “intracellular” under CC; and “protein binding”, and “ATP binding” under MF (Figure 2H). GO enrichment analysis results suggested that the “microtubule-based movement”, “catechol oxidase activity”, “serine-type endopeptidase inhibitor activity”, and “microtubule” were significantly enriched terms. Forty-seven pathways of the target genes of lncRNAs were identified on the basis of KEGG enrichment analysis, including “Phenylpropanoid biosynthesis”, “Spliceosome”, “Protein processing in endoplasmic reticulum”, “Starch and sucrose metabolism”, “RNA transport”, “Carbon metabolism”, “Plant-pathogen interaction”, “Biosynthesis of amino acids”, and “Endocytosis” pathways (Figure 2I).

### 3.3. Target Gene Prediction of lncRNAs

To better explore the functions of the lncRNAs association with vernalization, we performed the prediction of the cis-target genes. The co-expression network of lncRNAs and mRNAs were uncovered by exploring the potential cis-regulatory functions of lncRNAs. One significant interaction between mRNAs and lncRNAs was uncovered that belonged to one-to-many interactions (Figure 3). One lncRNA, *MSTRG.26204.1*, and three mRNAs, *Bv8_189980_mizi.t1*, *Bv8_189990_perh.t1* and *Bv8_189991z13909_ihku.t1*, of the co-expression network may be essential for the vernalization of sugar beet. *Bv8_189980_mizi.t1* gene encoded a B3 domain-containing transcription repressor VAL1-like may be the vernalization gene of sugar beet. We dissected the expression patterns of genes in the co-expression network at time point (i.e., vernalization at 0, 2, 4, 6, 8 and 10 weeks) (Figure 4). Moreover, the secondary structures of *MSTRG.26204.1* were further investigated by RNAfold (http://www.unafold.org (accessed on 18 August 2022)), and the results were very similar to the *COOLAIR* (HG975389) of *A. thaliana* (Appendix A).

Moreover, we further investigated the differentially expressed mRNAs based on the results of the co-expression network and the encoding domain. Interestingly, we uncovered the AP2/ERF and B3 domain-containing transcription factor RAV1 downregulation in response to cold treatment. In our results, 34 genes encoding B3 domain-containing proteins were achieved, of which most were upregulated by vernalization such as B3 domain-containing transcription factor VRN1, VRN1-like, VAL1 and VAL2. Those genes positively responding to cold treatment were also identified in wheat and *A. thaliana*, indicating that they were possibly the vernalization genes of sugar beet. We also identified the FT-interacting protein 1 that was downregulated by cold treatment. To better understand the functions, we further analyzed the relationships of evolution and the expression patterns of these genes (Figure 5; Figure 6 qRT-PCR).

### 3.4. Verification of Vernalization-Related mRNAs and lncRNAs by qRT-PCR

We analyzed the results of RNA-Seq and selected 19 specifically expressed mRNAs and 10 lncRNAs between the vernalized and nonvernalized samples to verify the RNA-Seq results by qRT-PCR. The RNA-Seq results were confirmed reliability in the light of the expression patterns of those RNAs. The 29 (19 mRNAs and 10 lncRNAs) specifically expressed RNAs at different time point (i.e., vernalization at 0, 2, 4, 6, 8 and 10 weeks) were further analyzed by qRT-PCR (Figure 7A,B). The expression trends of mRNAs and lncRNAs suggested that they were positive in response to vernalization in sugar beet, and their expression levels were upregulated with prolongation of cold treatment.

## 4. Discussion

Vernalization contributed to multiple physiological processes of plant development including bolting, the initiation of reproductive growth and regulation of flowering time. With the advances in genomic and bioinformatic techniques, lncRNAs are emergent and becoming more and more important for plant development. It has been previously reported that lncRNAs involved in multiple plant development, such as lateral root development, vernalization, photomorphogenesis, pollen development, fiber development and nodulation [31].

The molecular mechanism of vernalization in *A. thaliana* has been almost fundamentally uncovered; however, the molecular mechanism of vernalization in *B. vulgaris* is still in its infancy. Here, we obtained 570 differentially expressed mRNAs and 292 lncRNAs by whole-transcriptome analysis between vernalized and nonvernalized sugar beet. We identified the first vernalization-related mRNAs and lncRNAs genes to lay foundation for further investigation of the vernalization in sugar beet. GO and KEGG analysis were conducted to explore the potential functions of differentially expressed mRNAs and the targets of lncRNAs. Our study obtained some significant GO terms that were association with vernalization, including histone H3-K36 methylation, animal organ formation, response to low fluence blue light stimulus by blue low-fluence system, histone methyltransferase activity (H3-K4 specific), chromosome, nuclear envelope, cell differentiation, ATP-dependent helicase activity, DNA binding, heterotrimeric G-protein complex, mRNA processing, DNA replication, flower development, DNA recombination, binding, negative regulation of flower development, protein binding, nucleus, helicase activity and nucleic acid binding, suggesting that these clusters may be involved in vernalization. Among those clusters, the histone modifications and chromatin-modifying complexes were enriched at floral repressor loci where they stably repressed floral repressor genes. Histone H3K36 methylation, as a epigenetic memory code was essential for *FLC* expression, involved in promotion of flowering after cold treatment [32,33,34]. It was also revealed that H3K4me played widespread roles in regulating *FLC* expression in plants [35].

The experimental results showed that eight significantly expressed mRNAs were enriched in circadian rhythm pathways and two significantly expressed mRNAs were enriched in flavone and flavonol biosynthesis pathways during vernalization (Appendix A), the results of which were concordant with previous studies [3,36]. Among the differentially expressed mRNAs, several regulatory proteins playing vital roles in the vernalization pathway were found, including three zinc finger CCCH domain-containing proteins, three F-box proteins, two flowering-time-regulated proteins FY and FPA, one PHD finger protein EHD3 (Appendix A). The CCCH zinc finger proteins were potential RNA-binding proteins family that related to mRNA stability. The CCCH zinc finger proteins participated in many developmental processes of plants, such as flowering, senescence and environmental responses [37,38,39,40].

Many factors interacted with one another in the whole life cycle of plants to promote optimum growth. F-box proteins, as a DNA-associated transcriptional factor, participated in multiple developmental processes of plants, including seed germination, root development, leaf development, floral development, and response to photoperiodism [41,42,43]. F-box proteins were located in both nucleus and membrane, where they regulated the C2H2-type transcription factor’s stability via the ubiquitin-26S proteasome pathway [44]. F-box transcripts also highly recruited in root and changed distribution in response to external conditions [45]. FY was essential for the regulation of FCA expression, and their interaction was associated with the function of the floral repressor *FLC* in floral transition. FPA located near the N terminus comprised three repeated RNA recognition motifs (RRM), whose function is almost similar to FCA-controlled flowering via regulating the recruitment of mRNA and encoding FLC [46,47]. Plant homeodomain (PHD) finger proteins participated in chromatin remodeling and transcriptional regulation existing in almost all eukaryotes. The PHD finger proteins linked to the PRC2 activity in vernalization and regulated the balance between floral promoting and repressing signals to determine the time of reproductive development in the end [25,48,49].

Transcriptional or post-transcriptional genes expression was regulated by lncRNAs in a cis-regulated manner [50]. A novel finding in this study was that one specific expressed lncRNA, with its target genes co-expression network, may play crucial roles in sugar beet during vernalization (Figure 3). To better explore the function of the lncRNA, the secondary structure of the lncRNA was dissected (Appendix A). The specific ‘double stem-and-loop’ structure was constructed by a contribution of less than 100 nucleotides to the function of lncRNAs [51]. This significantly expressed lncRNA was very similar to *COOLAIR* on the structure that involved in transcriptional shutdown of floral regulator FLC of Arabidopsis during vernalization [52].

The differentially expressed gene *Bv8_189980_mizi.t1* encoded a B3 domain-containing transcription repressor VAL1-like, while the *Bv8_189990_perh.t1* gene has not been characterized. Additionally, 34 genes encoding B3 domain-containing proteins were identified in our results, among which 26 were upregulated by vernalization such as B3 domain-containing proteins VRN1, VRN1-like, VAL1 and VAL2. Those genes were considered as the vernalization genes in wheat and *A. thaliana*. We also identified the FT-interacting protein 1, which was downregulated by cold treatment. Furthermore, we analyzed the expression patterns of these genes at different time points (i.e., vernalization at 0, 2, 4, 6, 8 and 10 weeks) (Figure 7). Our results indicated that the expression patterns obviously reached their highest levels at a time point of 10 weeks in vernalization. Moreover, the differentially expressed *BvVAL1-like*, *BvVRN1*, *BvVRN1-like*, *BvVAL1* and *BvVAL2* genes all contained the B3 domains, indicating that they were perhaps the vernalization genes of sugar beet. In addition, *MSTRG.26204.1*, cis-regulating the *BvVAL1-like* expression, was the long noncoding RNA association with vernalization of sugar beet.

The B3 domain-containing protein is a plant-specific domain that universally exists in the plant kingdom. The B3-domain transcription factor VAL1 located in the nucleation region in vivo triggered Polycomb silencing at *FLC* of *A. thaliana* during vernalization [53,54,55]. *VAL1* also regulated the floral transition by repressing *FT* [56]. Arabidopsis *VRN1* gene mediated the vernalization process by encoding a MADS-box transcription factor binding to DNA in vitro in the cold-induced transformation of reproductive growth [57,58,59]. *VRN1* was also a vernalization gene in wheat (*Triticum aestivum*) playing a vital role in reproductive growth [60]. Moreover, studies revealed that *VRN1* was the homolog of APETALA1/FRUITFULL function on the upstream of *FT* transcripts and upregulating *FT* expression in wheat [61]. More than 100 B3 domain-containing proteins were explored in *A. thaliana*, and one of these was involved in an epigenetic process called vernalization and was essential for floral initiation after cold treatment. In addition, B3 DNA binding domains of the transcriptional repressors *VAL1* and *VAL2* mediated the epigenetic silencing machinery of the *FLC* by the VAL1-B3–DNA structure complex during vernalization [54]. With five domains, AtVAL1 was a multi-function protein that recruited the histone modifier PHD–PRC2 complex to the *FLC* nucleation region by the specific B3 domains [62]. The epigenetic silencing of the *FLC* gene was mediated by Polycomb group (PcG) proteins during vernalization. The epigenetic silencing initiation process of the *FLC* gene was as follows: B3 DNA binding domains of *VAL1* bound to an *FLC* intronic RY motif within the Polycomb group (PcG) proteins nucleation region to modulate PcG repression of FT during the floral initiation [56]. PRC2 was recruited to the target loci by the VAL1/VAL2-RY regulatory complex which comprised VIVIPAROUS1/ABI3-LIKE1 (VAL1) and VAL2 proteins [55]. A DNA-binding B3 domain protein was encoded by *AtVAL1* in Arabidopsis, which was very important for seed maturation and floral transition [63]. Overexpression of *VaRAV1* in grape cells enhanced its cold tolerance via regulating the expression of target genes involved in cell wall composition [64]. Further exploration of the functions of these vernalization-related mRNAs and lncRNAs will clearly illuminate the molecular mechanisms of vernalization in sugar beet.

## 5. Conclusions

In our research, whole-transcriptome sequencing was employed to study the vernalized and nonvernalized roots of sugar beet. A total of 570 mRNAs and 292 lncRNAs with significance were identified between the samples. GO annotation and KEGG enrichment analysis indicated that several regulatory proteins played important roles in vernalization, such as zinc finger CCCH domain-containing proteins, F-box proteins, flowering-time-regulated proteins FY and FPA, and PHD finger protein EHD3. Furthermore, we found a novel vernalization-related lncRNA–mRNA–target genes co-expression regulatory network in *B. vulgaris*, comprising three mRNAs, *BvVAL1-like*, *Bv8_189990_perh.t1* and *Bv8_189991z13909_ihku.t1*, and one lncRNA, *MSTRG.26204*. Moreover, the B3 domain-containing proteins VRN1, VRN1-like, VAL1 and VAL2 were possibly the vernalization genes involved in the response to cold treatment.

## Figures and Tables

**Figure 1 genes-13-02217-f001:**
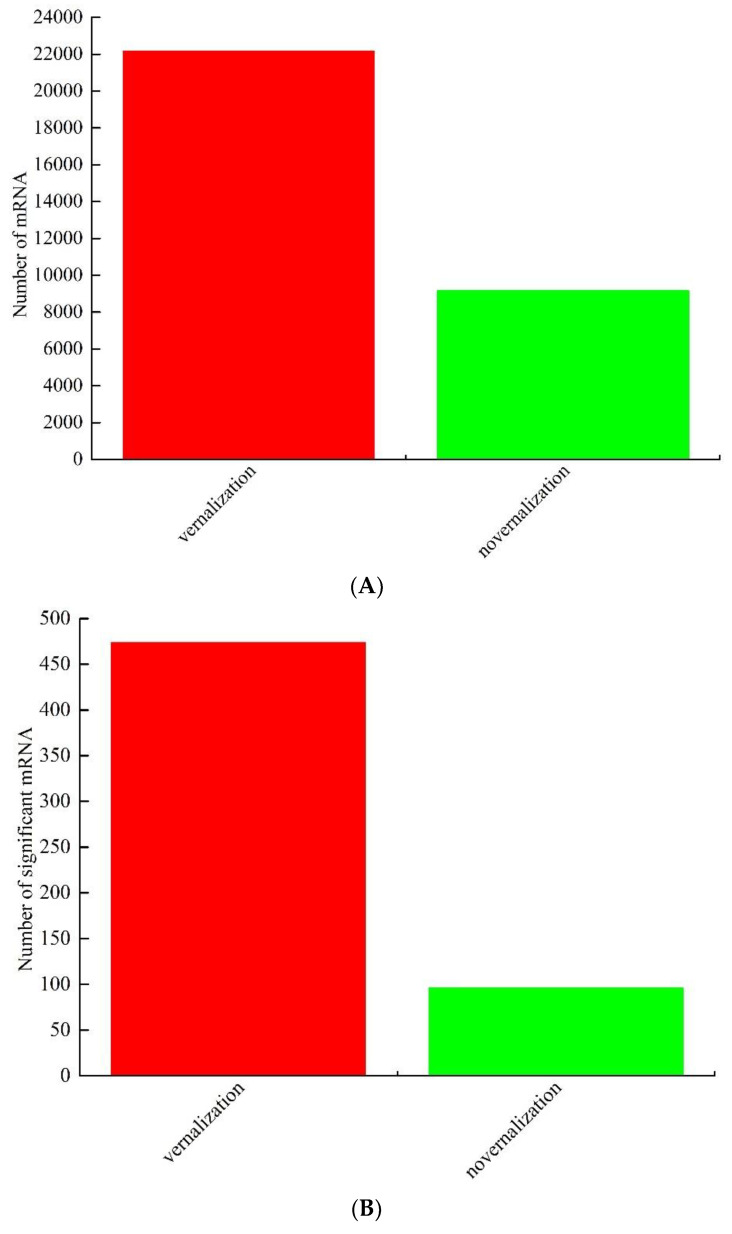
Identification and analysis of mRNAs under vernalization. (**A**) The number of differentially expressed mRNAs between the vernalized and nonvernalized samples; (**B**) The number of significantly expressed mRNAs between the vernalized and nonvernalized samples; (**C**) heat map; (**D**) Gene Ontology (GO) classifications; (**E**) GO enrichment analysis; and (**F**) Kyoto Encyclopedia of Genes and Genome (KEGG) pathway assignments for all of the differentially expressed mRNAs.

**Figure 2 genes-13-02217-f002:**
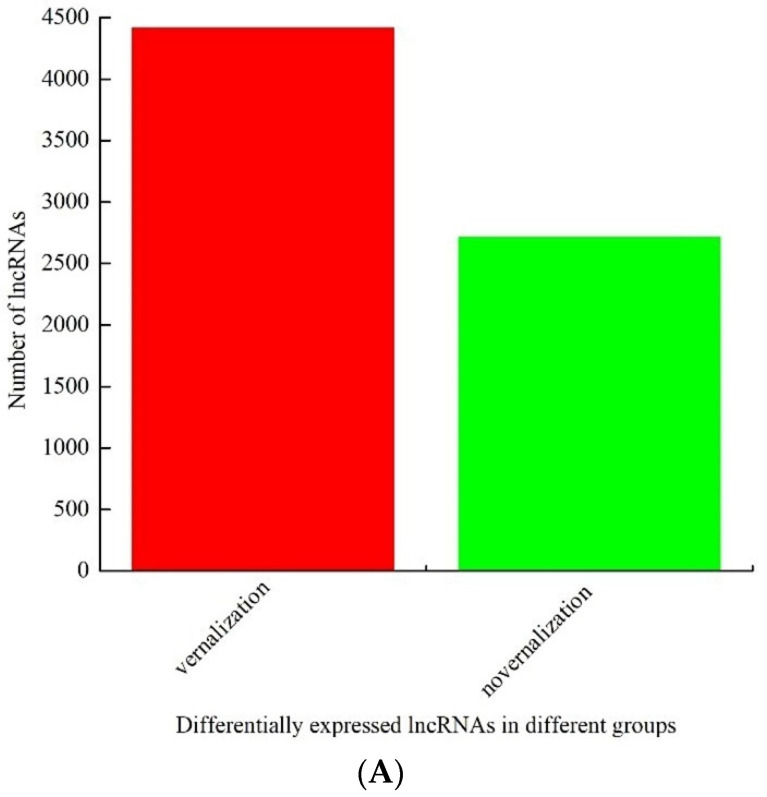
Identification and analysis of long-coding RNAs (lncRNAs) under vernalization. (**A**) The number of upregulated and downregulated differentially expressed lncRNAs identified between the vernalized and nonvernalized samples; (**B**) The number of significant differentially expressed lncRNAs between the vernalized and nonvernalized samples; (**C**) The volcano plot of differentially expressed lncRNAs; (**D**) Transcript length distribution of differentially expressed lncRNAs and mRNAs; (**E**,**F**) ORF length distribution of differentially expressed lncRNAs and mRNAs; (**G**) The exon number of differentially expressed lncRNAs and mRNAs; (**H**) Gene Ontology (GO) classifications; and (**I**) Kyoto Encyclopedia of Genes and Genome (KEGG) pathway assignments for all differentially expressed lncRNAs.

**Figure 3 genes-13-02217-f003:**
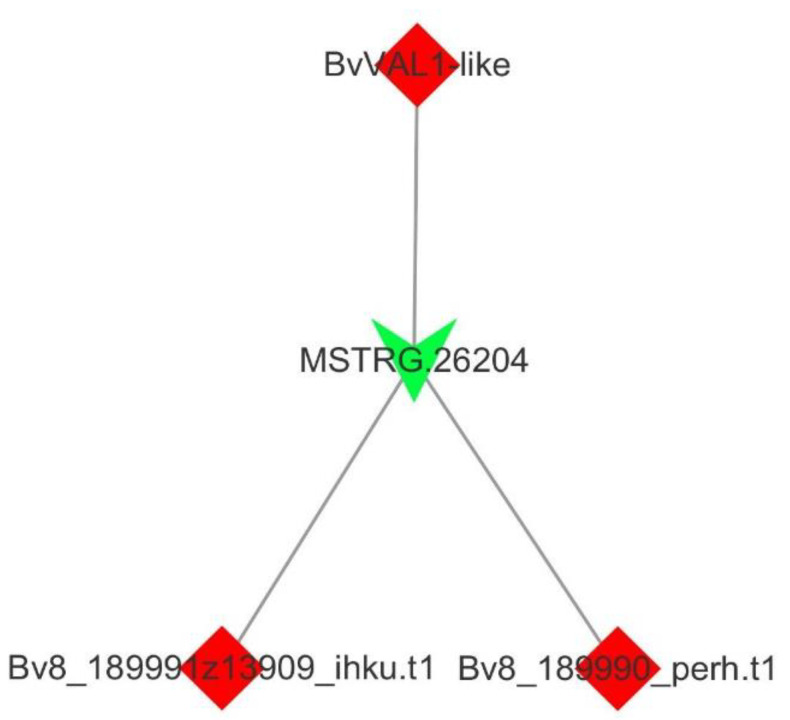
Co-expression network of differentially expressed long-coding RNAs and differentially expressed mRNAs. Green arrow indicate long noncoding RNA; red diamond represent mRNAs.

**Figure 4 genes-13-02217-f004:**
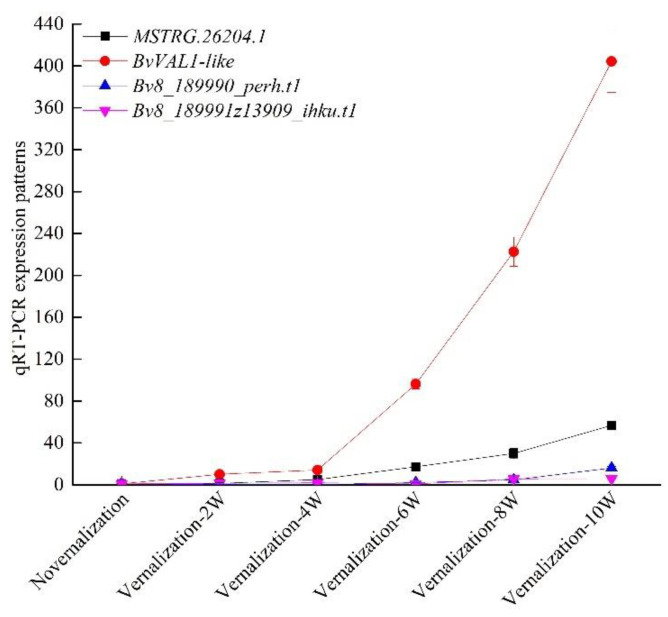
The expression patterns of long noncoding RNA (*MSTRG.26204*) and mRNAs (*Bv8_189980_mizi.t1*: *BvVAL-like*, *Bv8_189990_perh.t1* and *Bv8_189991z13909_ihku.t1*) (Appendix A) in the co-expression network of sugar beet.

**Figure 5 genes-13-02217-f005:**
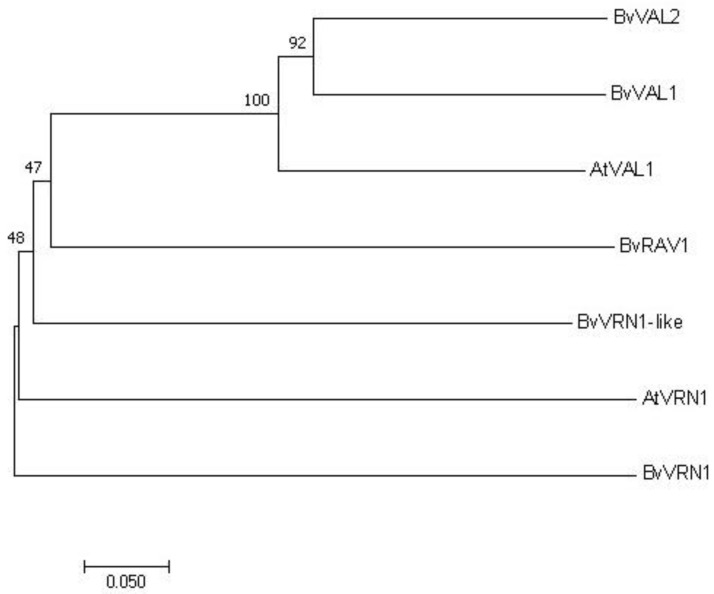
Phylogenetic tree of *BvVRN1-like* (Bv5_119850_xzuy.t1), *BvVRN1*(Bv6_136060_dknq.t1), *BvRAV1*(Bv6_140480_qrgc.t1), *BvVAL1*(Bv7_179010_ycqe.t1) and *BvVAL2*(Bv7_156080_dced.t1) with *AtVAL1*(AT2G30470) and *AtVRN1*(AF289051 and AF289052). The tree was built using the Neighbor-joining method, whose Bootstrap replication was 500, and Partial deletion was chosen, whose site coverage cutoff (%) was 50.

**Figure 6 genes-13-02217-f006:**
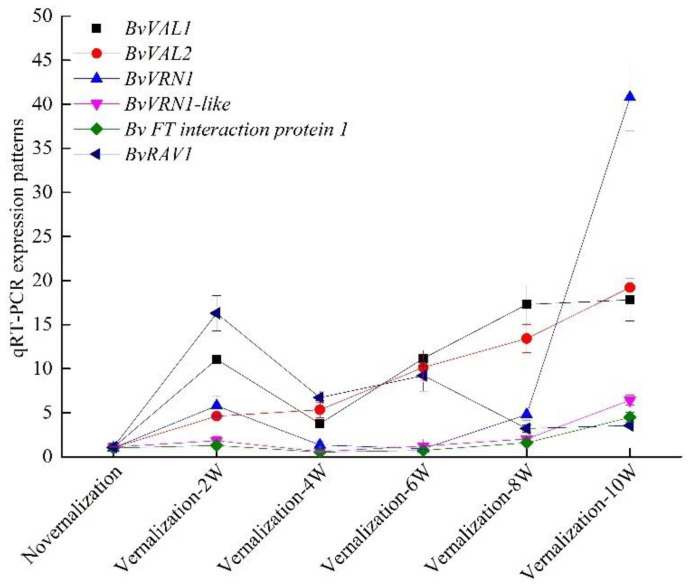
The expression patterns of *BvVRN1-like* (Bv5_119850_xzuy.t1), *BvVRN1* (Bv6_136060_dknq.t1), *BvRAV1* (Bv6_140480_qrgc.t1), *BvVAL1* (Bv7_179010_ycqe.t1), *BvVAL2* (Bv7_156080_dced.t1) and *BvFT-interacting protein 1* (Bv4_072340_puin.t1) (Appendix A).

**Figure 7 genes-13-02217-f007:**
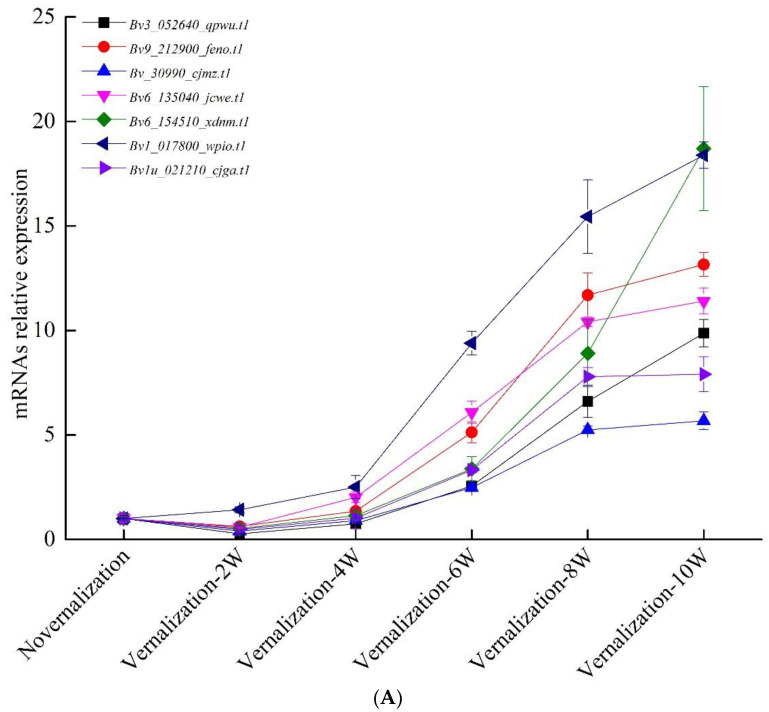
Quantitative real-time polymerase chain reaction (qRT-PCR) analysis at time point (i.e., vernalization at 0, 2, 4, 6, 8 and 10 weeks). The qRT-PCR results for (**A**) differentially expressed mRNAs; (**B**) differentially expressed long-coding RNAs. *BvICDH* was used as the internal control. Error bars, mean ± s.e. (*n* = 3).

## Data Availability

Not applicable.

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
