# Peer review of "Identification of the Genes Encoding B3 Domain-Containing Proteins Related to Vernalization of *Beta vulgaris"

_genes, 2022, doi:10.3390/genes13122217_

Round 1

Reviewer 1 Report

This research is functional and valuable work. 

Author Response

Dear Reviewer

Reviewer 2 Report

The manuscript is well written, the objective was well defined, as well as the analysis and discussion. On the first page of the results item, part of the material and methods was verified. Suggests including the text in the correct item. There was also a repetition of the table contents (5 and 6). You don't need that last sentence in the last paragraph. Figures 2D, 2E, 2F and 2G do not have clear captions. Some statistical analyzes (Phylogenetic tree) are not included in the material and methods item.

Suggestion: To justify why transcript above 200 bp was used. Generally, it recomends the selection of larger transcript for determination of genes or expressed regions (above 1000 bp).

Author Response

Dear Reviewer:

  please see the attachment. Thank you very much.

Reviewer 3 Report

In the present study, the comparison of the transcriptomic databases of vernalized and non-vernalized Beta vulgaris root samples was conducted. A total of 22159 differentially expressed mRNAs and 4418 differentially expressed lncRNAs were discovered.  For the first time, novel vernalization-related lncRNA-mRNA-target gene co-expression regulatory network and the candidate vernalization genes VRN1, VRN1-like, VAL1, and VAL2 encoding B3 domain-containing proteins were also unveiled. 

Overall, this is fairly good-written and conceived research, and minor corrections are needed. The manuscript needs moderate editing in the English language and style. Please find attached a pdf with integrated comments from the Reviewer. 

Author Response

Dear Reviewer:
